# Endosymbiont hijacking of acylcarnitines regulates insect vector fecundity by suppressing the viability of stored sperm

Brian L. Weiss[1]*, Fabian Gstöttenmayer[1], Erick Awuoche[1], Gretchen M. Smallenberger[1], Geoffrey M. Attardo[2], Francesca Scolari[3,4], Robert T. Koch[1], Daniel J. Bruzzese[1], Richard Echodu[5], Robert Opiro[5], Anna Malacrida[3], Adly M. M. Abd-Alla[6], Serap Aksoy[1]

1 Department of Epidemiology of Microbial Diseases, Yale School of Public Health, New Haven, Connecticut, United States of America, 2 Department of Entomology and Nematology, University of California, Davis, California, United States of America, 3 Department of Biology and Biotechnology, University of Pavia, Pavia, Italy, 4 Institute of Molecular Genetics, IGM CNR "Luigi Luca Cavalli-Sforza", Pavia, Italy, 5 Department of Biology, Gulu University, Gulu, Uganda, 6 Insect Pest Control Laboratory, Joint FAO/IAEA Centre of Nuclear Techniques in Food and Agriculture, Vienna, Austria

* brian.weiss@yale.edu

## Abstract

Competition between insects and their endosymbiotic bacteria for environmentally limited nutrients can compromise the fitness of both organisms. Tsetse flies, the vectors of pathogenic African trypanosomes, harbor a species and population-specific consortium of vertically transmitted endosymbiotic bacteria that range on the functional spectrum from mutualistic to parasitic. Tsetse's indigenous microbiota can include a member of the genus *Spiroplasma*, and infection with this bacterium causes fecundity-reducing phenotypes in the fly that include a prolonged gonotrophic cycle and a reduction in the motility of stored spermatozoa post-copulation. Herein we demonstrate that *Spiroplasma* and tsetse spermatozoa compete for fly-derived acyl-carnitines, which in other bacteria and animals are used to maintain cell membranes and produce energy. The fat body of mated female flies increases acylcarnitine production in response to infection with *Spiroplasma*. Additionally, their spermathecae (sperm storage organs), and likely the sperm within, up-regulate expression of *carnitine O-palmitoyltransferase-1*, which is indicative of increased acylcarnitine metabolism and thus increased energy demand and energy production in this organ. These compensatory measures are insufficient to rescue the motility defect of spermatozoa stored in the spermathecae of *Spiroplasma*-infected females and thus results in reduced fly fecundity. Tsetse's taxonomically simple and highly tractable indigenous microbiota make the fly an efficient model system for studying the biological processes that facilitate the maintenance of bacterial endosymbioses, and how these relationships impact conserved mechanisms (mammalian spermatozoa also use acylcarnitines as an energy source) that regulated animal host fecundity. In the case of insect pests and vectors, a better understanding of the metabolic mechanisms that underlie these associations can lead to the development of novel control strategies.

**Data availability statement:** All relevant data are in the manuscript and its supporting information files.

**Funding:** Funding was generously provided by the NIH/NIAID (R21AI163969 to BLW and SA, RO1AI139525 and RO1AI158805 to SA), the Ambrose Monell Foundation, the Yale Institute for Global Health (Spark Award to BLW), Cariplo-Regione Lombardia (research grant 'IMPROVE' to FS), and the Joint FAO/IAEA Centre of Nuclear Techniques in Food and Agriculture, Insect Pest Control Subprograms under the CRP D42017 (Agreement No. 26225) for the Research Project entitled 'Reproductive Biology of Glossina and Spiroplasma Effects'. These funding institutions had no role in the study design, data collection and analysis, decision to publish, or preparation of the manuscript.

**Competing interests:** The authors have declared that no competing interests exist.

## Author summary

Animals and the endosymbiotic bacteria that live inside them often compete for nutrients that both organisms require in order to survive. Tsetse flies, which transmit pathogenic African trypanosomes, can house several endosymbionts that have different impacts on their host's physiological well-being. Female tsetse flies that are infected with one of these bacteria, *Spiroplasma*, produce fewer offspring than do their uninfected counterparts. In this study we demonstrate that the bacterium and the fly's sperm cells (spermatozoa) compete for a specific type of lipid called acylcarnitines. When mated female tsetse flies are experimentally manipulated to produce less acylcarnitine *Spiroplasma* density decreases. Additionally, spermatozoa stored in the sperm storage organs of acylcarnitine depleted females lose motility and are eventually resorbed, thus rendering the females reproductively sterile. These findings mechanistically demonstrate how endosymbiotic bacteria can manipulate their host's reproductive potential. In the case of arthropods that transmit pathogenic microbes, this relationship has significant implications for disease transmission and epidemiology.

## Introduction

Spermatozoa are one of the most metabolically active cell types in the animal kingdom. This characteristic largely reflects the conspicuous beating of the tail that is necessary for sperm cells to successfully navigate the female reproductive tract and fertilize the ovulated egg(s). Many female animals, prominent among them insects, store spermatozoa in specialized organs after copulation. Stored spermatozoa are subsequently used to fertilize eggs that ovulate during consecutive reproductive cycles [1–3]. Most insect spermatozoa studied to date do not contain intrinsic energy reserves, and male seminal fluid-derived nutrients fail to remain active for long enough to sustain stored spermatozoa [4–7]. Thus, stored spermatozoa are nourished by the female. Meeting the metabolic requirements necessary to assure that stored spermatozoa are viable for prolonged periods of time presents a conspicuous metabolic burden for the female.

Tsetse flies (*Glossina* spp.) employ a specialized reproductive process called 'adenotrophic viviparity', during which females ovulate one egg per gonotrophic cycle (GC) and the subsequent embryo and larva are sustained in the maternal uterus throughout development [8]. Following copulation female tsetse flies store spermatozoa in specialized organs called 'spermathecae'. Stored sperm cells are released from the spermathecae to fertilize eggs as they ovulate during consecutive GCs. Female tsetse exhibit little or no reproductive senescence and can produce 8–10 progeny over the course of their approximately 3–4 month life span [9,10]. Although some tsetse species are polyandrous [11,12], its currently unknown how late in life these flies will continue to mate and whether spermatozoa from the first or subsequent matings are responsible for fertilization. However, viable sperm cells have been

observed in tsetse's spermathecae for more than 200 days post-copulation [13], indicative of the fact that female's must invest significant metabolic resources to sustain the spermatozoa for prolonged periods of time. To date nothing is known about the physiological mechanisms that impact long term sperm storage in tsetse's spermathecae.

Tsetse adults are cyclic vectors of old-world trypanosomes, which cause socioeconomically and epidemiologically devastating human and animal African trypanosomiases in sub-Saharan Africa. While only a small percentage of tsetse are infected with trypanosomes (~10–15%) [14], these flies all house a population-dependent assortment of viruses and transient and maternally-transmitted endosymbiotic bacteria [15]. With regard to the latter group of microbes, all tsetse flies harbor the obligate mutualist *Wigglesworthia*. This bacterium produces B-vitamins that are found in insufficient quantities in tsetse's vertebrate blood specific diet but that are required for the fly to produce energy via the tricarboxylic acid cycle [16–19]. In the absence of *Wigglesworthia*, pregnant female tsetse abort their larva because they lack the energy required to produce enough milk to complete larvigenesis [20,21]. In addition to *Wigglesworthia*, some tsetse flies are also infected with commensal *Sodalis* and parasitic *Wolbachia*, the latter of which can induce strong cytoplasmic incompatibility [22].

Some populations of tsetse species within the Palpalis subgenus can also house parasitic *Spiroplasma*. This bacterium's distribution and symbiotic relationship with wild tsetse flies has been most well studied in populations of *G. fuscipes fuscipes* (hereafter designated '*Gff*') from northern and western Uganda. In the northwest Albert Nile watershed approximately 34% of *Gff* house *Spiroplasma glossinidia* (hereafter designated strain '*sGff*'), while an average of 12.5%, 5.5%, and 3% of *Gff* collected in northcentral (the Achwa and Okole River watersheds), northeast (Lake Kyoga watershed), and western (Kafu River watershed) Uganda, respectively, harbored the bacterium [23]. *sGff* resides within tsetse's gut, reproductive tract, and hemolymph [24,25] [likely both intra- and extracellularly as does *Spiroplasma* in other insects [26–28]], and infection with this maternally-transmitted bacterium has a profound impact on the fly's physiology. Both laboratory reared and field captured *Gff* that house *sGff* are more refractory to infection with trypanosomes than are their counterparts that lack the bacterium [23,29]. The physiological mechanism(s) that underlies this *sGff*-induced phenotype is currently unknown. Additionally, the GC of *sGff*-positive (*sGff*+) *Gff* females is significantly prolonged, likely because of competition between the female fly and the bacterium for triacylglycerides [*sGff* which cannot synthesize lipids *de novo* [30,31]] that make up a significant component of tsetse milk [32,33]. Finally, spermatozoa that originate from *sGff* males exhibit a motility defect following transfer to the female spermathecae [33]. All of these outcomes reflect competition between tsetse and *sGff* for a finite supply of metabolically critical nutrients.

In this study we investigate how *sGff* manipulates tsetse's metabolism, with a focus on how the bacterium and spermatozoa compete for specific lipids that both cell types require to maintain their metabolic homeostasis. Our results further define the basic biological mechanisms that mediate tsetse fly-*sGff* symbiotic interactions and provide insight into factors that mediate long term sperm storage and thus population structure of this important vector of pathogens that cause disease in humans and domesticated animals. Knowledge obtained from this study is likely applicable to other arthropod pest model systems that house this bacterium as well as other parasitic endosymbionts.

## Results

### The *sGff* infection status of female *Gff*, but not males, impacts the motility of stored spermatozoa

We investigated whether the *sGff* infection status of mated *Gff* female's correlates with the motility of stored spermatozoa. To do so we quantified the beat frequency of sperm stored within the spermathecae of *sGff*+ and *sGff*-negative (*sGff*-) *Gff* females at 72 hr and 14 d post-copulation. We observed that spermatozoa stored in the spermathecae of *sGff*+ females beat at a significantly lower frequency than did spermatozoa stored in the spermathecae of *sGff*- females at both 72 hr and 14 d timepoints (72 hr, *sGff*+ = 12.5 ± 1.2 Hz, *sGff*- = 17.9 ± 1.3 Hz; 14 d, *sGff*+ = 7.4 ± 0.6 Hz, *sGff*- = 19.6 ± 2.1 Hz; Fig 1A) (S1 and S2 Video files, spermatozoa liberated from the spermathecae of *sGff*- and *sGff*+ females, respectively). Additionally, by 14 d post-copulation, the beat frequency of spermatozoa stored in the spermathecae of *sGff*+ females had decreased

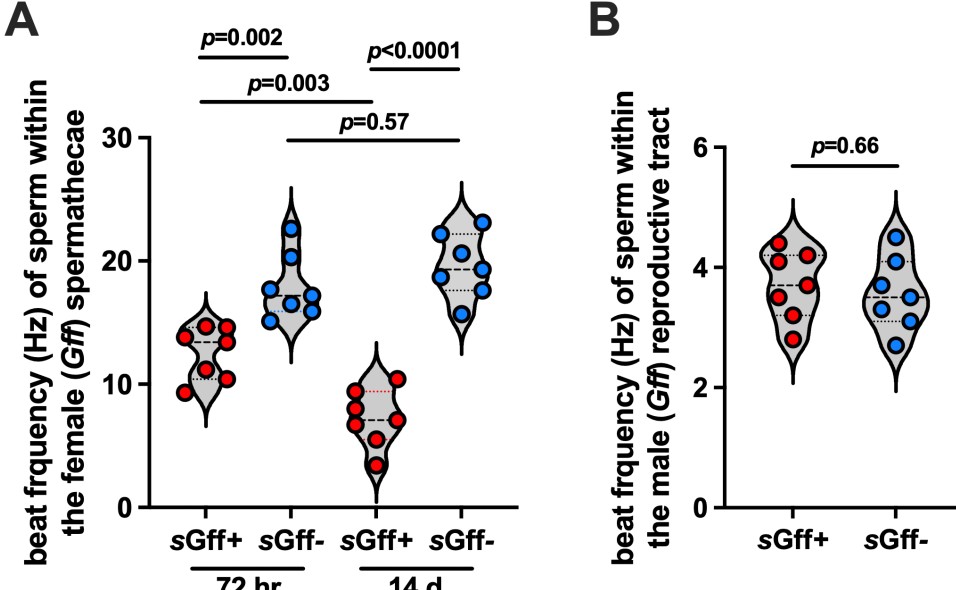

**Fig 1. The motility of sperm stored in the spermathecae of *s*Gff⁺ females is compromised.** Motility, as a measure of flagellar beat frequency in hertz (Hz), was quantified by acquiring video recordings of sperm tails at a rate of 30 frames per second. Beat frequency was analyzed using FIJI and the ImageJ plugin SpermQ. (A) *s*Gff⁺ and *s*Gff⁻ *Gff* females were mated with *Spi*⁻ males, and 72 hrs and 14 days later their spermathecae were removed and teased open with a fine needle to allow spermatozoa to exude. (B) Forty-eight hrs post-copulation, testes were removed from *s*Gff⁺ and *s*Gff⁻ *Gff* males and teased open with a fine needle to allow spermatozoa to exude. Points on graphs (A) and (B) represent one biological replicate (one replicate equals the average beat frequency of two sperm tails from one spermatheca or testis). Bars represent median values, and statistical significance was determined via one-way ANOVA with Tukey's multiple comparisons (GraphPad Prism v.10.4.1).

significantly (from $12.5 \pm 1.2$ Hz at 24 hr to $7.4 \pm 0.6$ Hz at 7d), while the beat frequency of spermatozoa stored in the spermathecae of *s*Gff⁻ females did not change over the same period of time (Fig 1A). These findings indicate that when a female fly is infected with *s*Gff, spermatozoa stored in her spermathecae exhibit reduced motility, and this phenotype becomes significantly more pronounced over time (between 72 hr and 14 d).

We previously demonstrated that sperm originating from *s*Gff⁺ males exhibited a motility defect following transfer to and storage within (24 hr post-copulation) the female spermathecae [33]. In light of our finding herein that the female's *s*Gff infection status impacts the motility of stored spermatozoa, we revisited the correlation between the male's infection status and this sperm phenotype. We observed that sperm in the male reproductive tract beat at a similar frequency regardless of whether or not the bacterium was present (*s*Gff⁺ = $3.7 \pm 0.32$ Hz, *s*Gff⁻ = $3.6 \pm 0.14$ Hz; Fig 1B). Furthermore, the beat frequency of spermatozoa in the testes of *s*Gff⁺ and *s*Gff⁻ males was significantly slower than that of spermatozoa stored in the spermathecae of *s*Gff⁺ and *s*Gff⁻ females, respectively (S1 Fig). Taken together these findings indicate that 1) spermatozoa in the testes of male *Gff* exhibit naturally low motility prior to ejaculation, as previously described in other insect systems [4,34], 2) spermatozoa motility increase following transfer to the female, and 3) the female's *Spiroplasma* infection status, as opposed to the male's, regulates spermatozoa motility following transfer into the spermatheca.

### *s*Gff infection alters the fat body lipidome of mated female *G. f. fuscipes*

*s*Gff⁺ females present phenotypes, including a lengthened gonotrophic cycle [33], suppressed motility of stored spermatozoa [herein and [33]], and increased refractoriness to infection with African trypanosomes [23,25], that are consistent with a metabolic imbalance within the fly. To determine the mechanism(s) that underlie these observations we performed an

untargeted LCMS lipidomics analysis of fat body tissue collected from mated sGff+ or sGff- Gff females one week following insemination (these flies were thus 10 days old). We chose to specifically identify and quantify the lipids produced by tsetse's fat body (which is the primary source of lipids in the fly) because these molecules play a prominent and well characterized role in maintaining tsetse's reproductive homeostasis [8,35]. We identified 1068 lipids from 28 lipid subclasses (S1 Dataset, tab 'fat body lipidomics_lab). Overall we observed that sGff- females presented with more fat body lipids than did their sGff+ counterparts (S2A Fig), and 23% of the fly fat body lipidome was altered when infected with sGff (S2B Fig). At the lipid subclass level, significant changes were observed in acylcarnitines, lysophosphatidylinositols, sphingomyelins, and sphingomyelin (phytosphingosines) (S2C Fig).

**sGff infection alters the abundance of acylcarnitines circulating in the hemolymph of mated female _G. f. fuscipes_.** Our data indicate that 1) spermatozoa stored in the spermathecae of sGff+ females are less motile than are their counterparts that reside in the spermathecae of sGff- females, and 2) despite the fact that overall sGff- females presented with more fat body lipids than did sGff+ individuals, we observed that mated Gff females that housed the bacterium presented with a relatively increased abundance of acylcarnitines in their fat body (S2C Fig). These findings are of particular interest because in mammals acylcarnitines serve as a prominent energy source that fuels metabolically demanding spermatozoa maturation and motility [36–39]. Acylcarnitines are polar, hydrophilic quaternary amines that arise from the enzymatic conjugation of activated fatty acids (_i.e._, fatty acyl-CoA) with L-carnitine. In animals acylcarnitines are a component of the 'carnitine shuttle', where they transport long chain fatty acids into the mitochondrial matrix for use in the production of ATP via fatty acid β-oxidation [40,41]. Mammalian acylcarnitines are produced in the male reproductive tract, specifically in the epididymis [38,42] where spermatozoa mature and are stored prior to ejaculation. Mammalian acylcarnitines are also produced in other tissues and transported into the male reproductive tract [38].The insect fat body [shown here in tsetse (S2C Fig) and in reference [43]] and gut [44] also produce these molecules to fuel energy intensive physiological processes such as nutrient metabolism and nutrient mobilization [44]. Acylcarnitines circulate in the insect hemolymph [45–47] where they are transported to other relatively nutrient intensive tissues such as flight muscles [44]. These molecules are also found in insect reproductive organs [48,49], although their origin and function therein have never been experimentally determined.

To investigate further the association between sGff infection status and acylcarnitine homeostasis we performed a targeted LCMS analysis to quantify the abundance of these molecules circulating in the hemolymph of mated sGff+ and sGff- Gff females. We found 26 acylcarnitine species circulating in the hemolymph of mated Gff females (S1 Dataset, tab 'hemolymph acylcarnitines_lab'; Table 1 lists the common names of all acylcarnitines identified in this study), of which seven [27%; AcCa(20:5), AcCa(18:1), AcCa(17:0), AcCa(16:0), AcCa(15:0), AcCa(10:2), and AcCa(7:0)] were significantly depleted in hemolymph collected from sGff+ individuals compared with that collected from individuals that were not infected with the bacterium (S3 Fig). Additionally, 16 (62%) more of the identified acylcarnitines were also depleted in the hemolymph of sGff+ compared to sGff- Gff females, although not significantly (Fig 2A and S1 Table).

Taken together our untargeted (fat body lipidome) and the targeted (circulating hemolymphatic acylcarnitines) lipidomics analyses revealed that sGff infection induces a change in acylcarnitine homeostasis in mated Gff females.

**sGff utilizes circulating acylcarnitines.** We observed that the fat body of mated, sGff+ females produces more acylcarnitines than does that of their sGff- counterparts (S2C Fig), but that acylcarnitines are significantly depleted from their hemolymph compared to that from mated sGff- Gff females. This finding suggests that hemolymph-borne sGff may be metabolizing acylcarnitines. This theory is supported by the fact that in addition to fueling metabolically intensive processes in animals, acylcarnitines are also of physiological importance to some bacteria in which they support growth [50], fuel motility [51], and serve as a source of free fatty acids to incorporate into their cell membrane [52]. To test this theory we used RNAi to perform a systemic knockdown of the gene that encodes Gff carnitine O-palmitoyltransferase-1 (CPT1; VectorBase gene ID GFUI030007; hereafter designated _Gffcpt1_) in four-day old mated sGff+ and sGff- Gff females. CPT1 is an enzymatic component of the carnitine shuttle that couples fatty acyl-CoAs to L-carnitine to form acylcarnitines.

**Table 1. Acylcarnitines identified in this study.**

| Common name | Fatty acid moiety |
| --- | --- |
| Acetylcarnitine | AcCa(2:0) |
| Hexanoylcarnitine | AcCa(6:0) |
| Heptanoylcarnitine | AcCa(7:0) |
| Octanoylcarnitine | AcCa(8:0) |
| Nonanoylcarnitine | AcCa(9:0) |
| Decanoylcarnitine | AcCa(10:0) |
| Decadienoylcarnitine | AcCa(10:2) |
| Dodecanoylcarnitine | AcCa(12:0) |
| Dodecanoylcarnitine | AcCa(12:1) |
| Dodecadienoylcarnitine | AcCa(12:2) |
| Tetradecenoylcarnitine | AcCa(14:0) |
| Tetradecenoylcarnitine | AcCa(14:1) |
| Pentaecanoylcarnitine | AcCa(15:0) |
| Palmitoylcarnitine | AcCa(16:0) |
| Palmitoleoylcarnitine | AcCa(16:1) |
| Heptadecenoylcarnitine | AcCa(17:0) |
| Heptadecenoylcarnitine | AcCa(17:1) |
| Stearoylcarnitine | AcCa(18:0) |
| Oleoylcarnitine | AcCa(18:1) |
| Octadecadienoylcarnitine | AcCa(18:2) |
| Linolenylcarnitine | AcCa(18:3) |
| Arachidylcarnitine | AcCa(20:0) |
| Eicoseneoylcarnitine | AcCa(20:1) |
| Eicosapenoylcarnitine | AcCa(20:5) |
| Docosadienoylcarnitine | AcCa(22:2) |
| Docosapentaenoylcarnitine | AcCa(22:5) |
| Docosahexaenoylcarnitine | AcCa(22:6) |
| Lignoceroylcarnitine | AcCa(24:2) |

The newly formed acylcarnitine molecules are then shuttled across the outer mitochondrial membrane towards the organelle's energy producing matrix. CPT1 abundance serves as an indicator of acylcarnitine formation and thus energy demand and production via the carnitine shuttle [40,53]. RNAi treatment (ds*cpt1*) reduced systemic *Gffcpt1* expression by approximately 70% (S4 Fig), which resulted in a significant depletion of 80% (16/20) of the acylcarnitines circulating in the hemolymph of treated individuals (S1 Dataset, tab 'cpt1_RNAi_lab'). We then quantified systemic (thorax and abdomen) *sGff* *16s rRNA* transcript abundance, as a reflection of bacterial load, in treatment and control (ds*gfp*) flies. We found that treatment with ds*cpt1* reduced s*Gff* abundance by an average of 48% when compared to s*Gff*+ females that received ds*gfp* control double-stranded RNA (Fig 2B). These findings imply that s*Gff* requires acylcarnitines to remain viable in the fly.

sGff can be cultured *in vitro* in BSK-H media supplemented with palmitic and oleic acids [see Materials and Methods and [30,54]]. These lipids comprise the fatty acid component of palmitoylcarnitine and oleoylcarnitine, respectively, both of which are highly abundant in the hemolymph of mated *Gff* females (S1 Dataset, tab 'hemolymph acylcarnitines_lab') and are significantly depleted in the hemolymph of s*Gff*+ compared to s*Gff*- individuals (Fig 2A). The two above-described findings led us further investigate the importance of these two acylcarnitines and their associated fatty acid moieties in

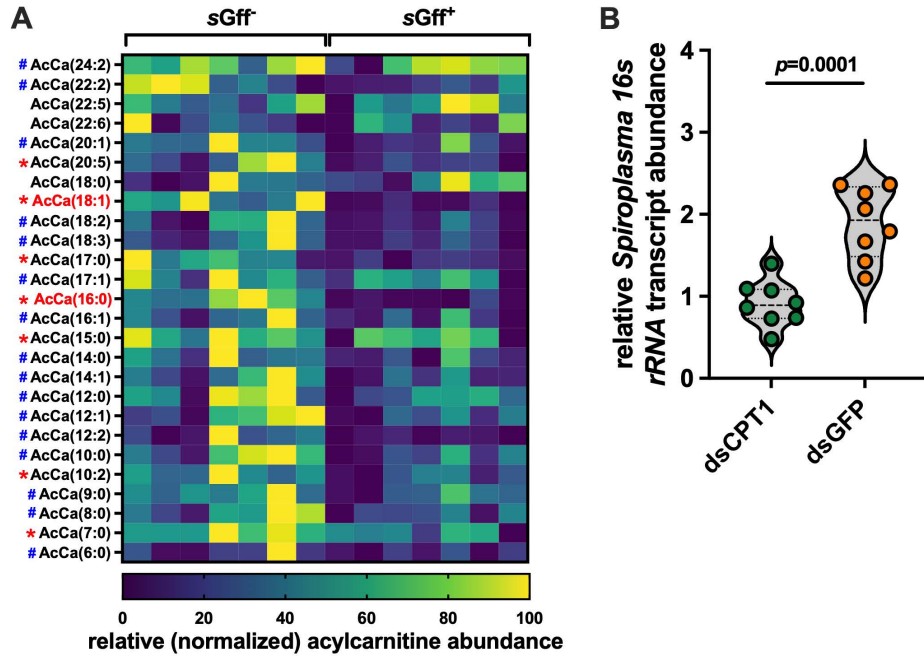

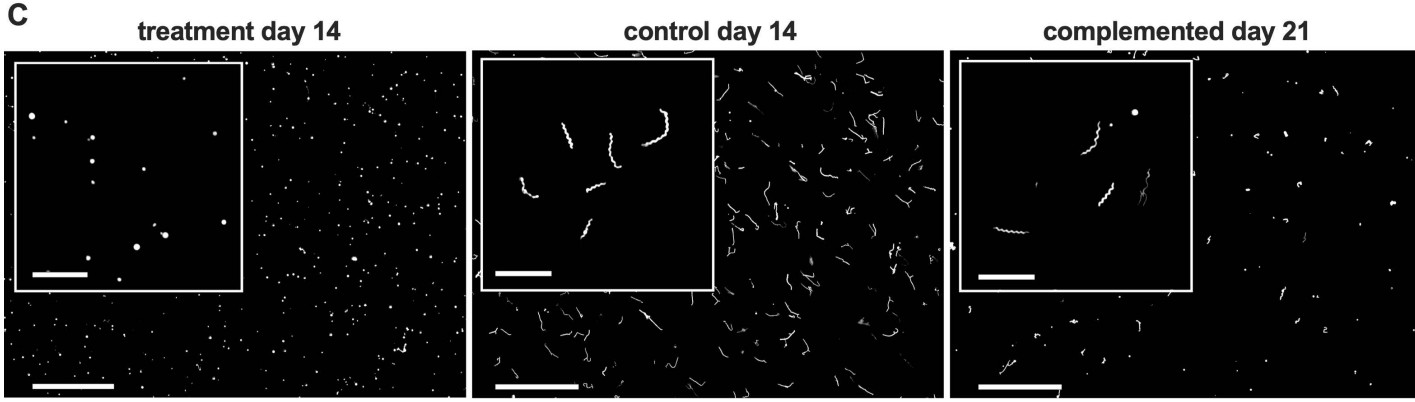

**Fig 2. *sGff* utilizes acylcarnitines to remain viable *in vivo* and *in vitro*.** (A) Acylcarnitine quantification in hemolymph collected from mated 10 day old mated *sGff⁺* and *sGff⁻ Gff* females (*n*= 7 biological replicates of each, each replicate containing 5 µl of hemolymph collected and pooled from two individual flies). Normalization of heat map data was performed by assigning a value of zero to the lowest output for each individual acylcarnitine, a value of 100 to the highest output for each individual acylcarnitine, and then normalizing all values between 0 and 100 accordingly. Red asterisks denote acyl-carnitines that are present at significantly higher ($p$<<0.05) quantities in the hemolymph of *sGff⁻* compared with *sGff⁺* individuals, while those denoted by a blue hashtag are also present at higher quantities in the hemolymph of *sGff⁻* individuals, although not significantly. Statistical significance was determined via multiple t-tests with false discovery rate (FDR) correction (GraphPad Prism v.10.4.1). Acylcarnitines denoted in red font are oleoylcarnitine [AcCa(18:1)] and palmitoylcarnitine [AcCa(16:0)]. (B) Relative systemic *sGff* density in mated *sGff⁺* females (14 days old, 10 days after treatment with dsRNA) following knockdown of *cpt1* expression via RNAi. Relative *sGff 16s rRNA* copy number was quantified from cDNA derived from the thorax and abdomen of treatment (*cpt1* dsRNA) and control (*GFP* dsRNA) flies. *sGff 16s rRNA* was normalized relative to *Gff gapdh* copy number in each sample. Each dot represents one biological replicate, and bars represent median values. Statistical significance was determined via Student's t-test (GraphPad Prism v.10.4.1). (C) Representative fluorescent micrographs of Syto9-stained *sGff* cultured in BSK-H media lacking palmitic and oleic acids (treatment day 14), complete BSK-H media (control day 14), and treatment cultures complemented with palmitic and oleic acids (complemented day 21). Scalebar of 40x photos = 50 µm, and scalebar of 100x insert photos = 10 µm.

sGff viability. We cultured the bacterium in media that lacked palmitic and oleic acids, and compared morphology of the cells with their counterparts grown in complete BSK-H media. We observed that sGff cultured in the absence of these fatty acids exhibit a stress phenotype. Specifically, when cultured for two weeks in BSK-H media lacking palmitic and oleic acid, the individual sGff cells present an abnormal coccus-like morphology (Fig 2C, left panel). Comparatively, sGff cells cultured for the same amount of time in complete BSK-H media present with a normal helical shape (Fig 2C, middle panel). To further confirm the importance of these fatty acids for sGff viability, we added palmitic an oleic acid back into the depleted media in which the abnormally spherical cells were maintained. This complementation partially rescued the stress phenotype such that many of the coccus shaped sGff returned to their normal helical morphology (Fig 2C, right panel). These results suggest that sGff requires acylcarnitines, possibly palmitoylcarnitine and oleoylcarnitine specifically (as sources of palmitic and oleic acid, respectively), to maintain their metabolic homeostasis and survive.

### Carnitine shuttle activity and acylcarnitine abundance is regulated by mating and sGff infection status in lab reared and field captured female *G. f. fuscipes*

Following copulation female tsetse flies store sperm within their spermathecae where they must remain viable for multiple gonotrophic cycles [11,13]. How these metabolically demanding cells are nourished within this environment remains unknown. Because acylcarnitines nourish spermatozoa in other animals [36–39], we theorized that they present a similar role in tsetse. To test this theory we first quantified *Gffcpt1* expression, which positively correlates with acylcarnitine production and energy demand [40,53], in the spermathecae of laboratory-reared virgin and mated sGff+ and sGff- *Gff* females two days post-mating (all individuals were five days old). We observed that *Gffcpt1* expression increased an average of 4.5-fold and 3.5-fold and in the spermathecae of sGff+ and sGff- *Gff* females, respectively, following mating (Fig 3A).

This outcome suggests that, regardless of sGff infection status, mating induces this tissue to ramp up energy production via the carnitine shuttle to provide nutritional support to resident spermatozoa. We next confirmed that sGff does reside in the spermathecae of sGff+ females (S5 Fig) and then monitored the longer-term impact of housing the bacterium in this tissue, and elsewhere in the fly, on spermathecal *cpt1* expression. We observed that *Gffcpt1* transcript abundance was significantly higher (average 3.1x) in spermathecae from older (14 days post-mating, all flies 17 days old) mated sGff+ compared to mated sGff- individuals (Fig 3B).

We also compared *Gffcpt1* expression in reproductive tracts of mated sGff+ and sGff- *Gff* females collected at two distinct field sites in northwest Uganda [Gorodona (GOR) and Oloyang (OLO), we did not know the specific age of these flies but confirmed that they were all pregnant via visual identification of an intrauterine larvae]. We found that *Gffcpt1* transcript abundance was also significantly higher in reproductive tracts from sGff+ individuals (GOR, 2.5x higher; OLO, 2.7x higher; Fig 3C). Additionally, we quantified the abundance of acylcarnitines in the hemolymph of mated sGff+ and sGff- *Gff* females collected in GOR and observed that of the 19 acylcarnitines identified, seven [37%; AcCa(18:1), AcCa(18:0), AcCa(16:0), AcCa(14:1), AcCa(8:0), AcCa(7:0), and AcCa(6:0)] were significantly more abundant in sGff- compared to sGff+ females, and nine (47%) more were also depleted in the hemolymph of sGff+ compared to sGff- *Gff* females, although not significantly (Fig 3D and S1 Dataset, tab 'hemolymph acylcarnitines_field, and S1 Table).

Taken together, these data suggest that the spermathecae (and likely the sperm stored within) of mated, lab reared and field captured sGff+ females exhibit increased carnitine shuttle activity, and thus produce more acylcarnitines to increase energy production, than do their counterparts that do not house the bacterium. This may reflect competition between sGff, pregnant females, and stored spermatozoa for acylcarnitines, all of which require these molecules to sustain their metabolism.

### Acylcarnitine homeostasis correlates with the viability of stored sperm, and thus fecundity, in pregnant female *G. morsitans morsitans*

We next investigated how acylcarnitine homeostasis in the spermatheca of mated female tsetse functionally correlates with the viability of stored sperm and fly fecundity. To do so we used *G. morsitans morsitans* (*Gmm*) as a proxy for *Gff*

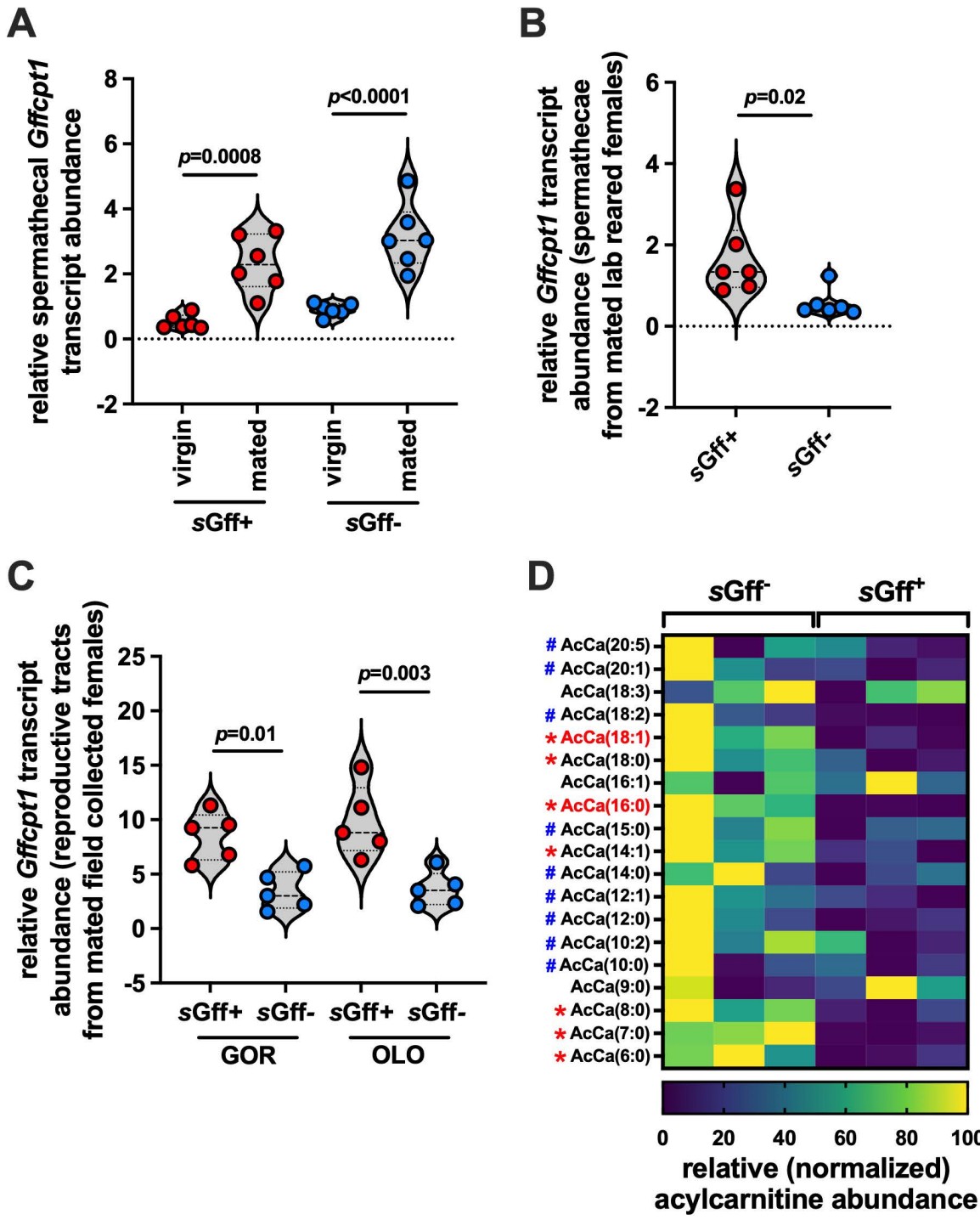

**Fig 3. Mating and *s*Gff infection status impact *cpt1* transcripts in the spermathecae of *Gff* females.** (A) Relative *Gffcpt1* transcript abundance in spermathecae extracted from five day old virgin and mated (two days post-mating) *s*Gff⁺ and *s*Gff⁻ *Gff* females. (B) Relative *Gffcpt1* transcript abundance in spermathecae extracted from mated *s*Gff⁺ and *s*Gff⁻ *Gff* females 14 days post-mating. (C) Relative *Gff cpt1* transcript abundance in reproductive tracts extracted from mated *s*Gff⁺ and *s*Gff⁻ *Gff* females collected at two distinct sites in Uganda (see Methods for more details about field collection flies). Each dot represents one biological replicate, with each replicate containing reproductive tracts collected and pooled from three individual flies. (D) Acylcarnitine quantification in hemolymph collected from mated, field collected *s*Gff⁺ and *s*Gff⁻ *Gff* females (*n* = 3 biological replicates of each, each replicate containing 5 µl of hemolymph collected and pooled from 20-25 individual flies). Normalization of heat map data was performed by assigning a value of

zero to the lowest output for each individual acylcarnitine, a value of 100 to the highest output for each individual acylcarnitine, and then normalizing all values between 0 and 100 accordingly. Red asterisks denote acylcarnitines that are present at significantly higher ($p < 0.05$) quantities in the hemolymph of sGff⁻ compared with sGff⁺ individuals, while those denoted by a blue hashtag are also present at higher quantities in the hemolymph of sGff⁻ individuals, although not significantly. Statistical significance was determined via multiple t-tests with FDR correction (GraphPad Prism v.10.4.1). Acylcarnitines denoted in red font are oleoylcarnitine [AcCa(18:1)] and palmitoylcarnitine [AcCa(16:0)]. In (A), (B), and (C) *Gff cpt1* was normalized relative to *Gff gapdh* copy number in each sample. In (A) and (B) each dot represents one biological replicate, with each replicate containing 10 spermathecae collected and pooled from five individual flies. Bars represent median values on all graphs. In (A) and (C) statistical analyses was determined via one-way ANOVA followed by Tukey's HSD post-hoc analysis, and in (B) statistical analyses was determined via Student's t-test (GraphPad Prism v.10.4.1.).

because *Gmm* does not harbor *Spiroplasma* [24], thus eliminating the need to screen for the presence of the bacterium in all individual flies following treatment. First, we compared *Gmmcpt1* expression and acylcarnitine abundance in spermathecae from 10 day old virgin versus mated *Gmm* females. This analysis revealed that the spermathecae of mated *Gmm* females expresses significantly more *cpt1* transcripts than does the organ from virgin females (S6 Fig). Additionally, the mated spermathecae produce five acylcarnitine species, four of which [AcCa(2:0), AcCa(16:0), AcCa(18:0), AcCa(18:1)] are found in significantly higher quantities in mated compared to virgin flies (Fig 4A and S1 Table). These findings indicate that mating upregulates spermathecal production of acylcarnitines, which implies that these lipids are metabolically important to stored spermatozoa, and thus fecundity, in mated *Gmm* females.

After confirming that tsetse's spermathecae produces acylcarnitines, and that their abundance increases significantly following mating, we investigated the impact of artificially reducing acylcarnitine abundance on the motility of spermatozoa in the male testes prior to mating, and in female spermathecae 7 days post-copulation. We supplemented the blood meal of 7 day old males (the age at which we normally use them for mating) and mated females 7 days-post mating (10 days old in total) with dsRNA that targets *Gmmcpt1* (VectorBase gene ID GMOY005222). The following day all flies received anti-*Gmmcpt1* dsRNA (ds*CPT1*) via thoracic microinjection. Age-matched control flies were similarly treated with anti-*GFP* dsRNA (ds*GFP*). *Gmmcpt1* expression was reduced by approximately 66% and 56% in spermathecae and testes of treatment compared to control females and males, respectively (S7 Fig). All dsRNA treated flies then received an additional normal blood meal, and 48 hrs later beat frequency of sperm located in male testes and female spermathecae was quantified. We observed no significant difference in beat frequency of sperm within the testes of ds*CPT1* treatment compared to ds*GFP* control males ($3.0 \pm 1.16$ Hz and $3.5 \pm 1.05$ Hz, respectively, $p = 0.3$; Fig 4B). This was expected, as insect spermatozoa are largely immotile in the male reproductive tract until immediately prior to ejaculation [55]. Conversely, spermatozoa within the spermathecae of ds*CPT1* treated females beat at a significantly lower frequency than did spermatozoa within the spermathecae of their ds*GFP* treated counterparts ($11.5 \pm 1.6$ Hz and $19.2 \pm 1.9$ Hz, respectively, $p < 0.0001$) (Fig 4C). We observed similar results when we inhibited CPT1 activity by treating flies with etomoxir, which specifically and irreversibly binds to CPT1 and represses its activity [56,57]. In this case spermatozoa within the spermathecae of etomoxir treated females beat at an average frequency of $12.4 \pm 1.9$ Hz, while spermatozoa within the spermathecae of control (PBS) females beat at an average frequency of $22.2 \pm 2.7$ Hz ($p = 0.0003$; S8A Fig). Taken together these data indicate that *Gmmcpt1* activity does not impact spermatozoa motility in male testes but does impact the motility of the cells following transfer to and storage within the female spermathecae.

We observed that sperm within the spermathecae of mated, ds*CPT1 Gmm* females exhibited compromised motility (similar to that of sperm within spermathecae of mated, *Spi⁺ Gff* females), and as such, we hypothesized this phenotype would result in reduced fecundity. To investigate this theory we supplemented the blood meal of *Gmm* females seven days-post mating with ds*CPT1*, and the following day we introduced ds*CPT1* into their hemocoel via thoracic microinjection. These flies then received another thoracic microinjection of anti-*Gmmcpt1* dsRNA one and three weeks later. Age-matched control flies were similarly treated with ds*GFP*. Fecundity, as measured by the number of pupae produced per female per gonotrophic cycle (GC) in treatment (ds*CPT1*) and control (ds*GFP*) groups, was then monitored over the course of four GCs. We observed that fecundity remained at similar levels in treatment (GC1 and GC2, 0.9 pupae/female)

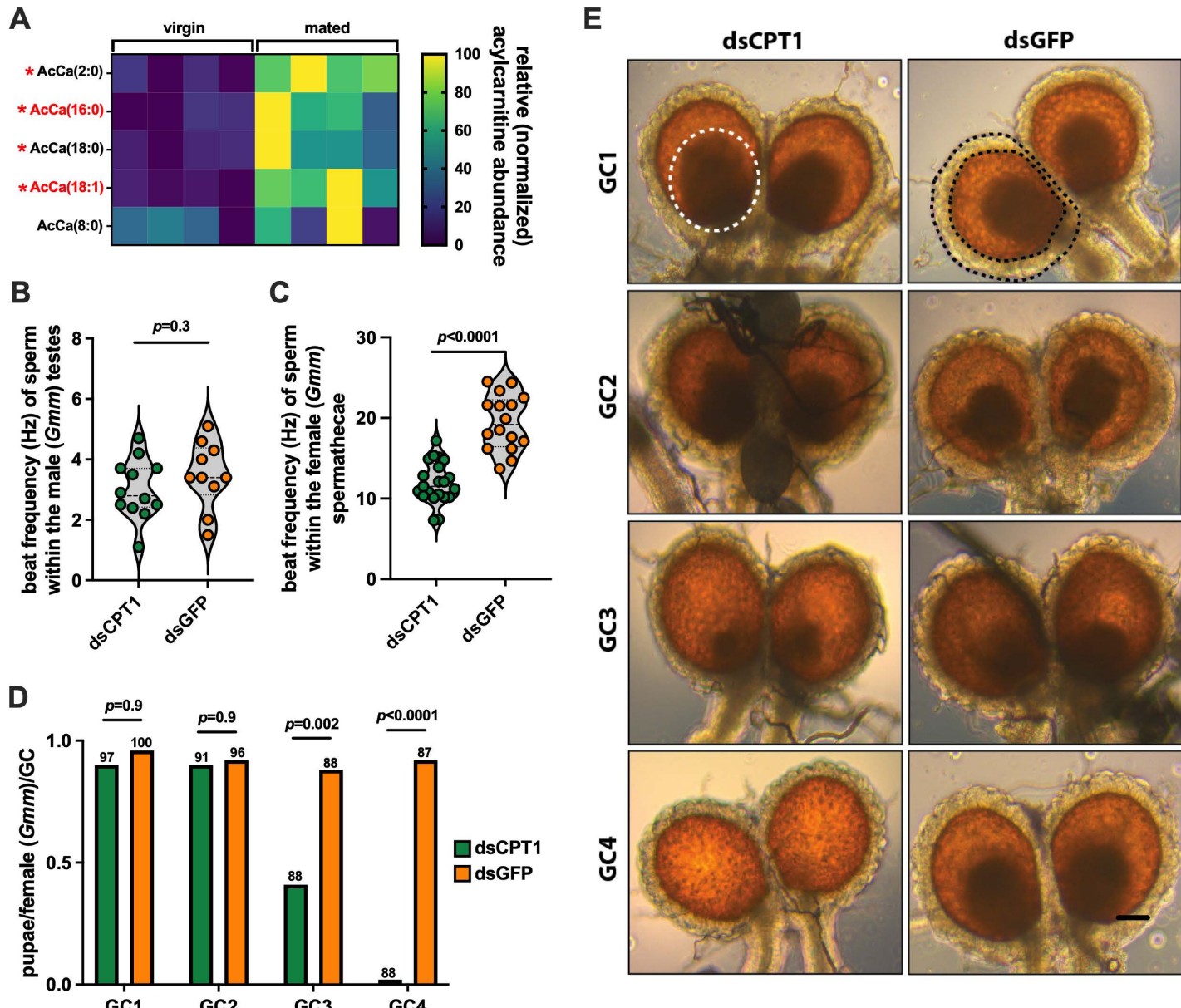

**Fig 4. Acylcarnitines are necessary for stored tsetse spermatozoa to remain viable and for the fly to remain fecund.** (A) Lipidomic analysis of spermathecae from virgin and mated *Gmm* females (*n* = 4 biological replicates of each, each replicate containing 25 spermathecae). Normalization of heat map data was performed by assigning a value of zero to the lowest output for each individual acylcarnitine, a value of 100 to the highest output for each individual acylcarnitine, and then normalizing all values between 0 and 100 accordingly. Red asterisks denote acylcarnitines that are present at significantly higher (*p* < 0.05) quantities in the spermathecae of mated compared to virgin *Gmm* females. Statistical significance was determined via multiple t-tests with FDR correction (GraphPad Prism v.10.4.1). Acylcarnitines denoted in red font are oleoylcarnitine [AcCa(18:1)] and palmitoylcarnitine [AcCa(16:0)]. (B and C) Motility, as a measure of flagellar beat frequency in hertz (Hz), of sperm within (B) male testes (one week old, mated) and (C) female spermathecae (10 day old, mated) from *Gmm* flies treated with either anti-*cpt1* (treatment, dsCPT1) or anti-*GFP* (control, dsGFP) dsRNA. Video recordings of sperm were acquired at a rate of 30 frames per second, and beat frequency was analyzed using FIJI and the ImageJ plugin SpermQ. Each dot on the graph represents one biological replicate (one replicate equals the average beat frequency of two sperm tails from one testis or one spermatheca). Statistical significance was determined via student's t-test using GraphPad Prism v.10.4.1. (D) Number of pupae deposited per individual per GC in distinct groups of mated, dsCPT1 or dsGFP treated *Gmm* females. Numbers above each bar represent pregnant female sample size at the end of each GC. Statistical significance was determined via multiple t-tests with FDR correction (GraphPad Prism v.10.4.1). (E) Representative micrographs of spermathecae extracted from mated, dsRNA treated *Gmm* females following the completion of GCs 1-4. The white dotted circle in the upper left panel (GC1, dsCPT1) highlights a sperm bundle, which is reduced in size in the spermathecae from dsCPT1 treated females following their 3rd GC (GC3) and absent in the spermathecae from dsCPT1 treated females following their 4th GC (GC4). The black dashes in the upper right panel (GC1, dsGFP)

highlight the glandular cells that surround the left spermatheca and presumably secrete acylcarnitines and other metabolites that nourish the sperm bundle stored within. Spermathecae from three dsCPT1 and dsGFP treated individuals, respectively, from each GC were examined to observe the status of their sperm bundle. Sperm bundle size was diminished in all three GC3, dsCPT1 treated individuals, and sperm were absent from the spermathecae of all three GC4, dsCPT1 treated individuals. Scale bar (GC4, dsGFP, lower right corner) = 250 μm.

and control (GC1, 0.96 pupae/female; GC2, 0.92 pupae/female) *Gmm* females over the course of two GCs. However, by the 3rd GC, fecundity had dropped significantly in treatment compared to control *Gmm* females (0.41 and 0.88 pupae/female, respectively), and by GC4, treatment females exhibited almost complete sterility (0.02 pupae/female compared to 0.92 pupae per control female) (Fig 4D). Interestingly, microscopic observation of spermathecae removed from treatment *Gmm* females revealed the presence of intact sperm bundles following the completion of GCs 1–3, although the sperm bundle present in the spermathecae from GC3 females was smaller. Conversely, no sperm bundles were present in spermathecae from treatment *Gmm* females following completion of the 4th GC (Fig 4E). Importantly, *Wigglesworthia* density was similar between dsCPT1 and dsGFP treated *Gmm* females at the end of GC4 (S9 Fig), which indicates that *Gmmcpt1* knockdown does not detrimentally impact this obligate endosymbiont and thus its role in facilitating tsetse reproduction. Conversely, this treatment did significantly repress *Sodalis* [this bacterium has no known function related to tsetse reproduction [15,58]], which, like *s*Gff, systemically infects tsetse [59] and may also consume acylcarnitines as an energy source. Instead, these findings suggest that in the presence of depleted *Gmmcpt1* expression, and thus reduced acylcarnitine production and reduced carnitine shuttle functionality, fecundity diminishes because stored sperm lack sufficient nutrients to remain viable for extended periods of time in the female spermathecae (S10 Fig).

## Discussion

Many insects house a stable population of endosymbiotic bacteria that perform numerous beneficial functions for their host, including nutrient provisioning, maintenance of immune system homeostasis, and direct protection against infection with pathogens. Conversely, some endosymbionts can present parasitic phenotypes, and in such cases the insect host must compete with these microbes for a limited supply of resources, which comes at a cost to both organisms [60]. Herein we demonstrate mated female *Gff* compete with endosymbiotic *s*Gff for metabolically critical acylcarnitines, which are lipids that nourish both the bacteria as well as spermatozoa that are stored long term within the female fly's spermathecae. When mated *Gff* females are infected with *s*Gff their fat body produces more acylcarnitine and their spermathecae upregulate *cpt1* expression and thus acylcarnitine formation. However, these compensative processes are insufficient and spermatozoa stored in the spermathecae of *s*Gff+ females present a motility defect. This results from the fact that acylcarnitines circulating in the fly's hemolymph, and those produced by the spermathecae, are depleted as the result of their utilization by *s*Gff (Fig 5). Additionally, although not experimentally demonstrated in this study, *s*Gff that reside in tsetse's gut presumably also utilize acylcarnitines. Such a cumulative impact on distinct fly tissues insinuates that the bacterium exerts a systemic effect on acylcarnitine homeostasis that impacts their availability locally in the mated female's spermatheca. Collectively, our results provide novel insights into the evolutionary mechanisms that facilitate the sustenance of endosymbioses and the metabolic mechanisms and symbiotic influences that underlie long term sperm storage and fecundity in female insects.

    *s*Gff's genome does not encode any of the molecular machinery required to synthesize lipids *de novo* [30,31], and as such the bacterium must acquire these molecules from its environment. This process is well documented with diacyl- and triacylglycerides, which *S. poulsonii* and *s*Gff obtain from the hemolymph of their insect hosts [33,61]. Our results herein provide two compelling lines of evidence indicating that *s*Gff also requires hemolymph-borne acylcarnitines in order to maintain its metabolic homeostasis. First, when acylcarnitine formation is experimentally inhibited by knocking down *Gff-cpt1* expression, *s*Gff abundance significantly decreases (as measured by *s*Gff 16s rRNA transcript abundance). Second,

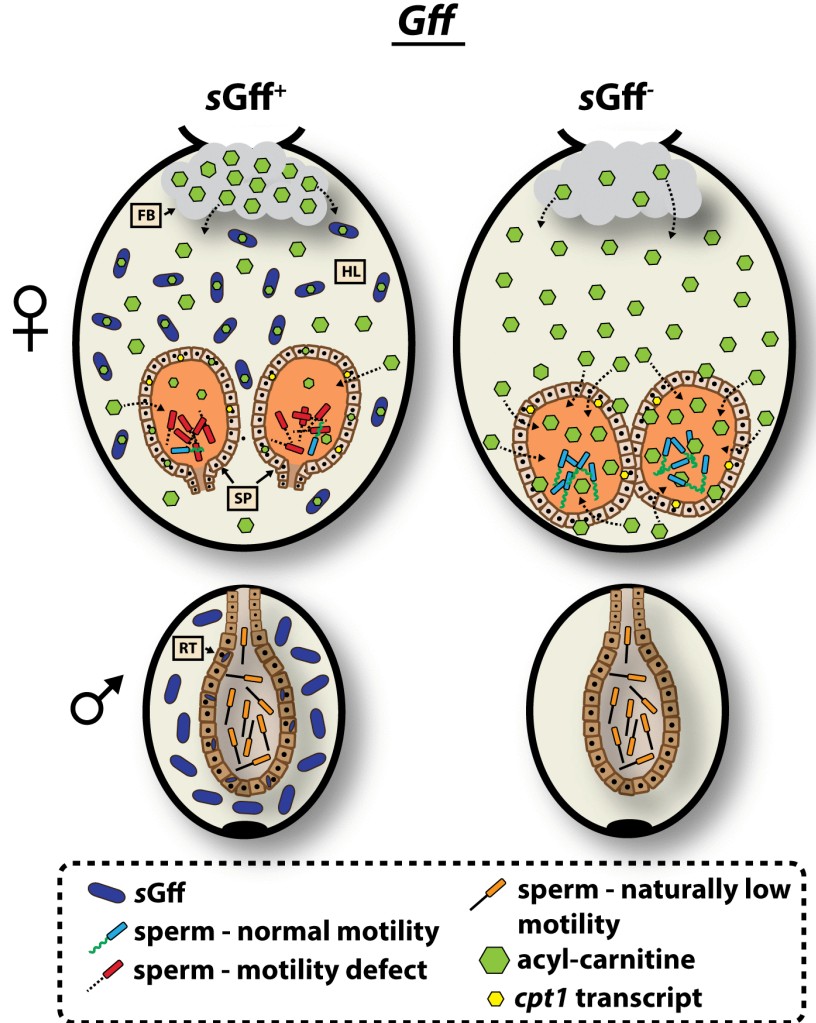

**Fig 5. Model illustrating the functional relationship that underlies *s*Gff and *Gff* competition for metabolically critical acylcarnitines.** Following mating, female tsetse flies store sperm in specialized organs called spermathecae. Sperm stored in the spermatheca of *s*Gff⁺ females presents a motility defect when compared with stored sperm in *s*Gff⁻ females. Tsetse spermatozoa and *s*Gff compete for fly derived acylcarnitines that both require to maintain their metabolic homeostasis. *s*Gff that reside in tsetse's hemolymph and spermathecae (and likely also other tissues) utilize acylcarnitines, which leaves fewer of these molecules circulating in the hemolymph for spermathecal uptake and utilization by stored sperm. In response, the fat body of mated *s*Gff⁺ females increases acylcarnitine production, and their spermathecae express a relative abundance (in comparison to that from mated *s*Gff⁻ females) of transcripts that encode carnitine O-palmitoyltransferase 1 (*cpt1*). Increased *cpt1* expression is indicative of increased acylcarnitine synthesis and enhanced energy production, via the carnitine shuttle, likely by both the spermathecae and sperm stored within. Despite these compensatory processes, spermatozoa stored in the spermathecae of *s*Gff⁺ females still lack enough acylcarnitine to sustain normal motility. Spermatozoa stored within the reproductive tract of *Gff* males naturally exhibits low motility regardless of *s*Gff infection status. FB, fat body; HL, hemolymph; SP, spermatheca; RT, reproductive tract.

we observed that 1) hemolymph from *s*Gff⁺ females is significantly depleted of acylcarnitine, specifically palmitoylcarnitine and oleoylcarnitine (these are the two most abundant acylcarnitines in the hemolymph of *s*Gff⁻ *Gff* females, and they exhibit the largest change in abundance when *s*Gff is present), compared to hemolymph from *s*Gff⁻ *Gff* females, and 2) palmitic and oleic acids, which are the fatty acid constituents of palmitoylcarnitine and oleoylcarnitine, respectively, must be added to the *s*Gff culture medium in order for the bacterium to grow *in vitro* [similar to *S. poulsonii*, [54]]. *Spiroplasma*

*floricola* and *S. citri* grown in culture take up palmitic and oleic acids for incorporation into their cell membrane [62,63]. Similarly, *s*Gff that reside within tsetse likely also require these fatty acids for the same purpose, and the bacterium may acquire them at least in part by metabolizing (via a currently unknown mechanism) palmitoylcarnitine and oleoylcarnitine present in the fly's hemolymph. Interestingly, Spiralin, which is the most abundant protein on *S. citri's* cell membrane [22% of the total cell membrane proteome, [64]], is heavily palmitoylated [65,66]. Palmitoylation may help anchor Spiralin to *Spiroplasma's* cell membrane [67], which would enhance the membrane's structural integrity such that the bacterium maintains its natural shape and remains viable *in vivo* and *in vitro* [65,67,68].

Mosquito [4], fruit fly [69], and honey bee [70] sperm stored post-copulation receive nourishment in the form of lipids, glycerophospholipids, and lipases that are secreted from clumps of exocrine secretory cells situated adjacent to the spermathecae (and connected via a duct) and/or by glandular cells that surround the spermathecal reservoir. Many of these same molecules can also be found circulating in insect hemolymph [33,71,45], from which they may be taken up by spermathecal cells and used to nourish stored spermatozoa. To date no information exists about the physiological mechanisms that underlie the maintenance of spermatozoa viability during their storage in the spermathecae of female tsetse flies. Interestingly, we observed glandular cells associated with the outer perimeter of tsetse's spermathecal reservoir (as seen in Fig 4E), the function of which we did not specifically investigate in this study. However, by virtue of the fact that these cells (as well as spermatozoa stored within) are the only metabolically active ones associated with tsetse's spermathecae [72,73], our data imply that they do generate acylcarnitines, and that these acylcarnitines are a likely source of energy for stored sperm. Specifically, distinct pools of spermatheca-derived cDNAs all contain *cpt1* transcripts, indicative of acylcarnitine production. Additionally, spermathecal *cpt1* transcript abundance, and the abundance of specific acylcarnitine species, both increase significantly in the spermathecae after mating. We were also able to experimentally induce the stored sperm motility defect in mated female *Gmm* (which does not house *Spiroplasma*) by knocking down expression of *Gmmcpt1*. Finally, we observed that the spermathecae (and also likely the sperm stored within) of mated, *s*Gff⁺ females ramps up the carnitine shuttle (as evidenced by increased *cpt1* transcript abundance) in an effort to compensate for the bacterium's utilization of circulating acylcarnitines. These collective data provide strong evidence that tsetse spermatozoa stored in the female spermathecae rely at least in part on these molecules to maintain their viability. The specific role that acylcarnitines play in maintaining the viability of sperm from other insects is largely unknown. However, acylcarnitines have long been known to serve as a prominent energy source for spermatozoa stored in the mammalian epididymis [36–38], suggesting that this function is evolutionarily conserved across animal taxa.

In addition to the limited availability of acylcarnitines as a nutrient source to fuel tsetse reproduction, decreased quantities of these lipids could further negatively impact spermatozoa stored in *s*Gff⁺ females by failing to protect them from oxidative stress. Spermatozoa are highly susceptible to reactive oxygen species (ROS) because of their limited ability to repair oxidative damage [69]. This physiological shortcoming, coupled with the fact that stored sperm are metabolically active in a confined space for relatively long periods of time [74], puts them in conspicuous danger of succumbing to oxidative damage. Carnitines, including acylcarnitines, present potent antioxidant properties [43]. More specifically, these lipids can directly scavenge and stabilize free radicals and/or regulate the bioactivity of enzymes involved in generating and counteracting ROS-mediated oxidative stress [75,76]. Our observation that spermatozoa stored in the spermathecae of *s*Gff⁺ females exhibit a motility defect, and are eventually completely resorbed, may be a consequence of limited acylcarnitine availability, which would not only starve spermatozoa of an essential metabolite but also leave them susceptible the toxic effects of excessive ROS activity.

*s*Gff infection further impedes tsetse fecundity by increasing the length of the fly's GC. This outcome is likely a reflection of the fact that *s*Gff⁺ females present with relatively low levels of circulating triacylglycerides [33], which account for a large proportion of the total lipids found in tsetse milk secretions that serve as the sole source of nourishment for developing intrauterine larva [32,35]. Circulating triacylglycerides could be depleted in *s*Gff⁺ females because the bacterium may be directly consuming those lipids and/or their precursor diacylglycerides [as occurs in the *Drosophila-Spiroplasma*

(MRSO strain) system, [61]] along with acylcarnitines as an energy source. Alternatively or additionally, the abundance of circulating triacylglycerides may be indirectly impacted by s*Gff's* utilization of acylcarnitines, as the latter molecules play a prominent role in fueling lipid mobilization from fat body tissues. This phenotype was demonstrated in the triatome bug *Rhodnius prolixus* when *cpt1* (called '*RhoPrCpt1*') expression was knocked down via RNAi. Under these circumstances the fat body of treated individuals contained almost twice as much triacylglyceride as did the fat body of control individuals, indicative of reduced lipid mobilization from the tissue [44]. Similarly, triacylglyceride buildup was observed in fat body tissues from *Drosophila* and the mosquito *Aedes aegypti* when the expression of *cpt1* orthologues was inhibited [77,78]. Interestingly, similar to the acylcarnitines, we also observed that mated s*Gff*+ females presented with a relative abundance of sphingomyelins in their fat body, suggesting that the presence of the bacterium also dysregulates the homeostasis of these lipids. Such an outcome could further inhibit tsetse fecundity, as sphingomyelins are another prominent component of tsetse milk [79]. In tsetse's fat body, lipolysis and lipid mobilization are tightly regulated by Brummer lipase and the adipokinetic hormone/adipokinetic hormone receptor, juvenile hormone, and insulin/IGF-like signaling pathways [35,80]. Future studies are needed to determine whether the lipid imbalance presented by mated, *Spi*+ females results from the bacterium's interference in one or more of these pathways.

The effects of s*Gff* induced inhibition of sperm motility, combined with the lengthening of tsetse's GC, would exert a profoundly negative pressure on tsetse populations that house the bacterium. This may account for why s*Gff* infection prevalence remains relatively low in certain wild populations of *Gff* (between 0% and 34%) and *G. tachinoides* (37.5%) [23,24] [notably, *Spiroplasma* DNA was recently detected in a low percentage (17 out of 1,136; 1.5%) of *G. pallidipes* collected in the Maasai Mara National Preserve, Kenya [81]]. Tsetse are K-strategists with a low rate of reproduction [8], so if *Spiroplasma* infection prevalence were to reach fixation or near fixation, the fly host population would likely collapse. Concordantly, the recent collapse of a *Gff* colony in Bratislava, Slovakia may at least in part reflect the fact that 80% of individuals were infected with s*Gff* [24].

Tsetse flies are the prominent vectors of pathogenic African trypanosomes (*Trypanosoma* spp.), which are the causative agents of epidemiologically and socioeconomically devastating human and animal African trypanosomiases in sub-Saharan Africa [82,83]. Vertebrate infective bloodstream form African trypanosomes contain high concentrations (1–5mM) of L-carnitine [84], and biochemical analyses of the mechanisms that underlie the production of ATP strongly imply that they employ the carnitine shuttle as a component of their energy production strategy [85]. Additionally, although never specifically examined experimentally, insect stage procyclic form parasites also produce ATP conventionally in their mitochondria [86], thus suggesting they too make use of the carnitine shuttle [84,87]. Our previous observation that s*Gff*+ *Gff* are more refractory to infection with trypanosomes than are *Spi*- individuals [23,25] may at least partially reflect bacterial (*i.e.*, s*Gff*) modification of lipid availability that comes at the metabolic expense of a cohabitating pathogen (*i.e.*, *T. b. brucei*). This type of competitive relationship is not unprecedented, and has been described in other insect models that house parasitic endosymbionts. For example, *Wolbachia* and *Drosophila* C virus competition for dietary cholesterol in *Drosophila* may account for why the fly presents increased resistance to the virus when it is infected with the bacterium [88]. Paredes et al. [89] demonstrated that *Drosophila* infected with *S. poulsonii* are more refractory to infection with parasitic wasp larvae than are uninfected flies because the bacterium consumes hemolymphatic diacylglycerols that the parasite also requires as a nutrient source. Another example of endosymbiont-pathogen competition for lipids occurs in the mosquito *Ae. aegypti*, which can vector several arboviruses, including dengue, West Nile, and Zika. In this case acylcarnitine-mediated virus replication is inhibited by *Wolbachia*, which diverts this energy source for its own metabolic needs [56]. Interestingly, competitive chemical inhibitors of the carnitine shuttle, such as bromoacetyl-L-carnitine and D-carnitine, have been examined as potential vertebrate trypanocidal chemotherapeutics [90,91]. Assuming insect stage African trypanosomes also rely on acylcarnitines for energy production, experimental simulation of s*Gff*-mediated acylcarnitine depletion in tsetse that do not harbor the bacterium could in theory 'starve' the parasites and inhibit their ability to complete their developmental program in the fly. This, in combination with s*Gff*-mediated suppression of tsetse population size (via reduced fecundity associated

with increased GC length and reduced sperm motility), could represent a multi-pronged approach to disease control. Importantly, similar disease control strategies could be employed in other model systems where the arthropod vector, indigenous bacteria, and parasites/pathogens rely on acylcarnitines as a nutrient source.

## Materials and methods

### Tsetse flies

*Gff* pupae were obtained from the Joint FAO/IAEA IPCL insectary in Seibersdorf, Austria and reared at the Yale School of Public Health insectary at 25°C with 70–80% relative humidity. *Gmm* were maintained in Yale's insectary at 25°C with 60–65% relative humidity. All flies housed in Yale's insectary were maintained under a 12 hr light:dark photo phase. Flies received defibrinated sheep blood every 48 hr through an artificial membrane feeding system [92].

 *Gff* field specimen were collected in the Fall of 2019 in the bush surrounding two villages, Gorodona (GOR; 3°15'57.6"N, 32°12'28.8"E) and Oloyang (OLO; 3°15'25.2"N, 32°13'08.4"E), in the Albert Nile watershed of northwest Uganda [for a map see Fig 1, reference [23]]. While all field captured flies were visually inspected to determine their mating status (individuals were identified as mated if spermatozoa were present), their precise age was unknown. Whole flies and hemolymph (collected as described in the '*Lipidomics sample collection*' subsection below) were flash frozen in liquid nitrogen, and subsequently shipped to the Yale School of Public Health on dry ice for analysis. To obtain reproductive tracts from these individuals, they were defrosted, the distal ends of their abdomens were quickly removed, and RNA was immediately processed (see subsection below entitled '*Determination of Glossina CPT1 and sGff 16s rRNA transcript abundance*' for relevant experimental details).

### Determination of *s*Gff infection status

Every *Gff* fly used in the experiments performed in this study was individually tested to determine its *s*Gff infection status. All adult female and male *Gff* were tested by extracting genomic DNA (gDNA) from two fly legs (lab reared specimen) or the terminal end of abdomens (field collected specimen), which contains the reproductive tract, using a DNeasy Blood and Tissue kit according to the manufacturer's (Qiagen) protocol. To confirm that intact gDNA was successfully extracted, all samples were subjected to PCR analysis using primers that specifically amplify *Gff tubulin*. *s*Gff infection status was determined by subjecting gDNA samples to PCR analysis using *s*Gff *16S rRNA* specific primers. All PCR primers used in this study are listed in S2 Table. The *Spiroplasma 16S rRNA* locus was amplified by PCR using previously published parameters [23]. Samples were considered infected with *s*Gff if the expected PCR product of 1000 bp was detected (see S11 Fig for an example gel).

 Detection of live *s*Gff within the spermathecae of mated *Gff* females was determined by RT-PCR. To remove potentially contaminating hemolymph-borne *s*Gff, dissected spermathecae were subjected to a gentamicin exclusion assay as described previously [93]. Purified, DNase treated RNA (200 ng), pooled from the spermathecae of three mated *s*Gff⁺ females, was reverse transcribed in cDNA, 200 ng of which was then subjected to PCR analysis using *s*Gff *16s rRNA* specific primers. The same cDNA pools were similarly subjected to PCR analysis using *Gff β-tubulin* specific primers.

### Lipidomics sample collection

The following samples were collected for lipidomics analyses:

1) Fat body from 10 day old mated, laboratory-reared *s*Gff⁺ and *s*Gff⁻ females (*n* = 7 biological replicates of each, each replicate containing two fat body tissues).

2) Hemolymph from 10 day old mated, laboratory-reared *s*Gff⁺ and *s*Gff⁻ females (*n* = 7 biological replicates of each, each replicate containing 5 µl of hemolymph collected and pooled from two individual flies).

3) Hemolymph from mated, field-captured *s*Gff⁺ and *s*Gff⁻ females (*n* = 3 biological replicates of each, each replicate containing 5 µl of hemolymph collected and pooled from 20-25 individual flies).

4) Hemolymph from dsCPT1 and dsGFP treated *Gff* females (*n* = 3 biological replicates of each, each replicate containing 5 µl of hemolymph collected and pooled from two individual flies). Flies were administered dsRNA as four day old adults, and hemolymph was collected 10 days later.

5) Spermathecae from five day old virgin and mated *Gmm* females (*n* = 4 biological replicates of each, each replicate containing 25 spermathecae).

Hemolymph was collected as described in [94]. All individual samples were collected on ice and initially stored separately at -80°C until the *s*Gff infection status of the flies from which they were derived was determined by PCR. Fat body and hemolymph samples from *s*Gff⁺ and *s*Gff⁻ flies were subsequently defrosted, pooled into the biological replicates described above, submerged in MeOH (fat body, 200 µl per replicate; hemolymph, 40 µl per replicate), and flash frozen. Pooled hemolymph from *s*Gff⁺ fly samples was centrifuged at 6000x *g* for 5 min (at 4°C) to remove *s*Gff cells.

## Untargeted fat body lipidomics

Fat body samples were topped off to 1 mL of MeOH and homogenized in bead beater vials prefilled with garnet pieces (Cole-Parmer, Metuchen, NJ, USA) in a TissueLyzer LT (Qiagen, Germantown, MD, USA) for 10 min at 50 Hz. The homogenates were transferred to 8 ml glass vials with an additional 1 ml MeOH and 4 ml chloroform. After vortexing, the samples were incubated for 10 min in an ultrasound bath. Two ml of water were added, and the samples were vortexed for 1 min. Phases were separated by centrifugation at 500xg at -5°C for 10 min. The chloroform phases were transferred to 8 ml glass vials and evaporated under nitrogen flow. The samples were finally resuspended in 60 µl of chloroform. Each sample was split into two equal aliquots, one for each polarity analysis.

LC-MS analyses were modified from Miraldi et al. [95] and performed on an Orbitrap Exactive plus MS in line with an Ultimate 3000 LC (Thermo Electron North America, Madison, WI, USA). Each sample was analyzed in positive and negative modes, in top 5 automatic data-dependent MS/MS mode with HCD fragmentation at stepped NCE 25,45. Column hardware consisted of a Biobond C4 column (4.6 × 50 mm, 5 µm, Dikma Technologies, Foothill Ranch, CA, USA). Flow rate was set to 100 µL min⁻¹ for 5 min with 0% mobile phase B (MB), then switched to 400 µL min⁻¹ for 50 min, with a linear gradient of MB from 20% to 100%. The column was then washed at 500 µL min⁻¹ for 8 min at 100% MB before being re-equilibrated for 7 min at 0% MB and 500 µL min⁻¹. For positive mode runs, buffers consisted for mobile phase A (MA) of 5 mM ammonium formate, 0.1% formic acid and 5% methanol in water, and for MB of 5 mM ammonium formate, 0.1% formic acid, 5% water, 35% methanol in isopropanol. For negative runs, buffers consisted for MA of 0.03% ammonium hydroxide, 5% methanol in water, and for MB of 0.03% ammonium hydroxide, 5% water, 35% methanol in isopropanol. Lipids were identified and their signal integrated using the Lipidsearch software (version 4.2.27, Mitsui Knowledge Industry, University of Tokyo). Integrations and peak quality were curated manually before exporting and analyzing the data in Microsoft excel.

## Targeted (acylcarnitines) hemolymph and spermathecae lipidomics

For targeted acylcarnitine analyses, hemolymph samples were topped off to 800 µl of MeOH, and proteins were precipitated by incubating the samples overnight at -20°C followed by centrifugation at 18,000xg at -9°C for 10 min. Supernatants were transferred to new microcentrifuge tubes and evaporated to dryness under nitrogen flow. Samples were resuspended in 15 µl of 1:1 MA:MB. Acylcarnitines were analyzed on either an Orbitrap Exactive plus MS in line with an Ultimate 3000 LC (laboratory-derived samples) or an ID-X MS in line with a Vanquish LC (field-derived samples) (both from Thermo Electron North America, Madison, WI, USA). Data was recorded in positive mode with a scan range of 140–1000

m/z in MS1 mode. Data dependent MS/MS analysis, with HCD fragmentation with stepped NCE 25,45, 50, was triggered on list of potential acylcarnitines ions generated by Lipidsearch software (version 4.2.27, Mitsui Knowledge Industry, University of Tokyo). Column hardware consisted of a Luna Omega polar C18 column (1.6 um 100x2.1mm, Phenomenex, Torrance, CA, USA). Flow rate was set to 300 µl min⁻¹. The gradient started with 2 min at 2% MB, then to 12% MB in 6 min, then to 100% MB in 4 min. After 4 min at 100% MB the column was re-equilibrated at 2% MB for 4min. MA consisted of water with 0.1% formic acid and MB consisted of Acetonitrile: isopropanol 9:1 with 0.1% Formic acid. A concentrated sample was used to identify the list and retention time of acylcarnitines present in the samples, by comparing with pure standards (for carnitine and acetyl carnitine) and by characteristic fragment ions (for the other acylcarnitines). The peaks corresponding to the acylcarnitines identified in the concentrated sample were integrated in each sample using xcalibur (Thermo Electron North America, Madison, WI, USA).

To identify and quantify acylcarnitines present in the spermathecae of virgin and mated *Gmm* females, the organs were dissected from five day old individuals. Upon completion of spermathecae collection, ice cold PBS was added to 100 µl and tissues were homogenized with a pestle, flash frozen, and stored at -80°C. Samples were shipped on dry ice to Metabolon (Morrisville, NC) and analysis was performed using the DiscoveryHD4 global metabolomics platform. Relative differences between samples were determined by comparison of mean metabolite abundance across four biological replicates, each containing 25 spermathecae.

### *s*Gff culture

Three high-density *s*Gff cultures [culture conditions are described in more detail in [30]] at different passage numbers (3, 7 and 10) were each split into new control or treatment vials and pelleted at 12,000 x g for 15 min. The supernatant was removed and cells were resuspended in either complete BSK-H media (control) or BSK-H media lacking palmitic and oleic acids (treatment). Upon observation of a change in morphology, treatment cultures were passaged into complete BSK-H media and observed again one week later. Cultures were monitored weekly by Syto9 staining (0.025 mM) and visualized on a Zeiss Axio Imager 2 fluorescent microscope equipped with an AxioCam MRm camera.

### RNAi and etomoxir mediated inhibition of *Gff* CPT1

Double stranded (ds) RNA used in this study was synthesized using a MEGAscript T7 *In Vitro* Transcription kit according to the manufacturer's (Invitrogen) protocol. dsRNA target specificity was confirmed *in silico* at VectorBase (www.vectorbase.org) via BLAST analysis against *Gff* and *Gmm* RNA-seq libraries, and a complete set of *Gff* and *Gmm* genomic scaffolds (also available on the VectorBase website). Additionally, all T7 RNA polymerase promoter-encoding PCR products were sequenced prior to the synthesis of dsRNA to confirm that they encoded only target genes.

To investigate the impact of suppressed *cpt1* expression on systemic *Spiroplasma* density, three day old, mated *Gff* females were administered their 2nd blood meal spiked with 1µg of either dsCPT1 or dsGFP RNA per 20µl of blood (the approximate amount a fly consumes per blood meal), and two days later the same females received a thoracic microinjection containing 1 µg of either dsCPT1 or dsGFP RNA. All flies were subsequently fed regular blood every other day for eight days. Three days after receiving their last bloodmeal (14 days old), treatment and control flies were sacrificed to determine their relative *s*Gff density. To investigate the impact of suppressed *cpt1* expression on sperm motility in the male testes, distinct groups of teneral (newly emerged, unfed adults) males were offered their first blood meal spiked with 1µg/20µl of blood of either dsCPT1 (treatment) or dsGFP (control) RNA. Two days later the same males received a thoracic microinjection containing 1 µg of either dsCPT1 or dsGFP RNA, and they were then exposed to an equal number of fertile females. All males were subsequently fed two more clean blood meals (males were 8 days old at this point), after which their testes were dissected and sperm housed within monitored to quantify their motility (see the 'Sperm motility assays' section below). To investigate the impact of suppressed *cpt1* expression on sperm motility in the female spermathecae post-copulation, distinct groups of teneral *Gmm* females were offered their first blood meal spiked with 1µg/20µl

of blood of either dsCPT1 or dsGFP RNA. Two days later the same females received a thoracic microinjection containing 1 µg of either dsCPT1 or dsGFP RNA, and they were then exposed to an equal number of fertile males. dsRNA treated, mated females received two more clean blood meals (females were 8 days old at this point), after which their spermathecae were dissected and sperm housed within monitored to quantify their motility (see the 'Sperm motility assays' section below).

Chemical inhibition of CPT1 was achieved through the use of etomoxir, which irreversibly binds and inhibits the enzyme's activity [56,57]. To determine etomoxir's toxicity towards tsetse we performed a 10 day LD 50 assay by thoracically micro-injecting distinct groups of pregnant female flies with either 25 µM, 75 µM, 125 µM, or 175 µM doses of the inhibitor [these doses were chosen based on experiments performed using *Drosophila* [96]]. Ten days post-treatment, extrapolation from our survival curve indicated that administering an etomoxir dose of 90 µM would kill 50% of flies (S8B Fig), and as such, we used 50 µM doses in subsequent experiments to quantify the impact of etomoxir-mediated CPT1 inhibition on the motility of stored tsetse sperm. Distinct groups of teneral *Gmm* females were fed once and injected the following day with either etomoxir (50 µM) or PBS, which was used to formulate the etomoxir suspension. The next day all females were mated, and then given two more bloodmeals. Spermathecae were dissected from individuals of both groups (8 days old at this point) and sperm housed within were monitored to quantify their motility (see the 'Sperm motility assays' section below).

### Determination of *Glossina* CPT1 and *s*Gff 16s *rRNA* transcript abundance

Reverse-transcription quantitative PCR (RT-qPCR) was used to quantify relative *Gff* and *Gmm cpt1*, and *s*Gff *16s rRNA* transcript abundance in treatment (*s*Gff⁺, dsCPT1 treated *Gff*) compared to control (*s*Gff⁺, dsGFP treated *Gff*) individuals. To quantify *Glossina cpt1* transcript abundance total RNA was extracted (using Trizol reagent) from 1) spermathecae from lab-reared *Gff* and *Gmm* females, 2) testes from lab-reared *Gmm* males, and 3) the distal end of abdomens (which contains the reproductive tract, larvae were removed from all individuals prior to RNA extraction), from field-captured *Gff* females (fly ages and sample sizes are indicated in corresponding figure legends). To quantify *s*Gff *16s rRNA* transcript abundance, total RNA was extracted from the thorax and abdomen of 14 day old mated, lab-reared *s*Gff⁺ females. All RNA samples were DNase treated (Turbo DNase, Invitrogen), and RNA quality and quantity was quantified using a NanoDrop 2000c (Thermo Scientific). 200 ng of total RNA was then reverse transcribed into cDNA using the iScript cDNA synthesis kit (BioRad), and cDNA was subjected to RT-qPCR analysis (two technical replicates were used for each sample). Relative expression (RE) was measured as $RE = 2^{-ddCT}$, and normalization was performed using the constitutively expressed *Glossina gapdh* gene. All PCR primers used in this study are listed in S2 Table. RT-qPCR was performed on a T1000 PCR detection system (Bio-Rad, Hercules, CA) under the following conditions: 8 min at 95°C; 40 cycles of 15 s at 95°C, 30 s at 57°C or 55°C, 30 s at 72°C; 1 min at 95°C; 1 min at 55°C and 30 s from 55°C to 95°C. Each 10 µl reaction contained 5 µl of iTaq Universal SYBR Green Supermix (Bio-Rad), 1 µl cDNA, 2 µl primer pair mix (10 µM), and 2 µl nuclease-free $H_2O$.

### Fecundity measurements

**Sperm motility assays.** Testes and spermathecae were excised from treatment and control flies and placed into 10 µl of HEPES-buffered saline solution [145 mM NaCl, 4 mM KCl, 1 mM MgCl2, 1.3 mM CaCl2, 5 mM D-glucose, 10 mM 4-(2-hydroxyethyl)-1-piperazineethane- sulfonic acid (HEPES), pH 7.4] in the well of a concave glass microscope slide. Testes and spermathecae were gently teased apart with a fine needle and sperm were allowed to exude from the organs for 1 minute. Sperm beating was recorded using an inverted microscope (10x phase contrast, Zeiss Primovert) that housed a charge-coupled camera (Zeiss Axiocam ERc 5s). Two sperm tails per sample were analyzed. Recordings were acquired at a rate of 30 frames per second, and beat frequency was analyzed using FIJI and the SpermQ ImageJ plugin [97].

**Wigglesworthia and *Sodalis* density in *Gmmcpt1* knockdown flies.** RT-qPCR was used to quantify the relative transcript abundance of the *Wigglesworthia* (*Wgm*) *recA* and *Sodalis* (*Sgm*) *rplB* genes in midguts collected from

mated *Gmm* females treated with either anti-*cpt1* (treatment) or anti-*gfp* (control) dsRNAs. The relative expression of these genes was used to indirectly determine whether RNAi treatment detrimentally impacted bacterial viability. These experiments were performed using procedures described in the '*Determination of Glossina CPT1 and Spiroplasma 16s rRNA transcript abundance*' subsection provided above.

**Pupae per female.** Distinct groups of teneral *Gmm* females were treated with either dsCPT1 or dsGFP RNA as described in the 'RNA interference' section above. However, instead of sacrificing the eight-day old mated females (as was done above to quantify sperm motility), they received two more intrathoracic microinjections of their respective dsRNAs at subsequent 10-day intervals while being maintained over the course of four GCs. At the conclusion of each GC, pupae were counted and the number of pupae per female was determined (dead females were counted at the conclusion of each GC to ensure accurate determination of pupae per female). At the conclusion of the 4th GC, several remaining treatment and control females were sacrificed to confirm the effectiveness of RNA interference and to microscopically monitor the status of sperm within corresponding spermathecae.

## Sample sizes and statistical analyses

All sample sizes are provided in corresponding figure legends or are indicated graphically as points on dot plots. Biological replication implies distinct groups of flies were treated in the course of consecutive experimental procedures. All statistical analyses were carried out using GraphPad Prism (v.10.4.1). All statistical tests used, and statistical significance between treatments, and treatments and controls, are indicated on the figures or in their corresponding legends.

## Supporting information

**S1 Video. Video, captured at a rate of 30 frames/second, of spermatozoa released from the spermatheca of *s*Gff⁻ *Gff* females.**
(MOV)

**S2 Video. Video, captured at a rate of 30 frames/second, of spermatozoa released from the spermatheca of *s*Gff⁺ females.**
(MOV)

**S1 Dataset. Lipidomics data from experiments performed in this study.**
(XLSX)

**S1 Table. A summary of the relative abundance of acylcarnitines in tisues of mated sGff+ and sGff- female Gff (sGff infection), and virgin and mated Gmm (mating).**
(XLSX)

**S2 Table. PCR primers used in this study.**
(XLSX)

**S1 Fig. Comparison of sperm beat frequency in male testes vs female spermathecae in mated *s*Gff⁺ and *s*Gff⁻ *Gff* males and females.** Regardless of *s*Gff infection status, spermatozoa in the male testes beat at a slower frequency than do spermatozoa stored in the female spermatheca. These data are the same as that depicted on Fig 1A and 1B, but combined into one graph to facilitate the comparison of sperm beat frequency in the male testes and the female spermathecae.
(TIF)

**S2 Fig. Overview of lipidome profiles of the fat body from mated *s*Gff⁺ and *s*Gff⁻ *Gff* females.** (A) 1110 lipids from 28 lipid subclasses were identified in fat body tissues from 10 day old mated *s*Gff⁺ and *s*Gff⁻ *Gff* females. Overall, fat body tissues from mated, *s*Gff⁻ individuals contained more lipids than did their *s*Gff⁺ counterparts. (B) Twenty-three percent of

the mated *Gff* female fat body lipidome changed significantly when individuals were infected with *s*Gff. (C) The abundance of lipids within four subclasses, the acyl-carnitines, lysophosphatidylinositols, sphingomyelins, and sphingomyelin phyto-sphingosines, changed significantly in the fat body of mated *s*Gff⁺ compared to *s*Gff⁻ *Gff* females.
(TIF)

**S3 Fig. Percent change of acyl-carnitine abundance in hemolymph of mated *s*Gff⁺ versus *s*Gff⁻ *Gff* females.** The abundance of 27% of the acyl-carnitines circulating in the hemolymph of mated *Gff* females changed significantly when individuals were infected with *s*Gff.
(TIF)

**S4 Fig. Efficiency of RNAi mediated knockdown of systemic *cpt1* expression in mated *Gff* females.** Relative systemic *Gffcpt1* transcript abundance in dsCPT1 (treatment) and dsGFP (control) treated (10 day old mated) *s*Gff⁺ females. RNAi reduced systemic *cpt1* expression by 70%. Each dot on the graph represents one biological replicate (each replicate including one thorax and abdomen from one dsRNA treated individual). Statistical significance was determined via student's t-test (GraphPad Prism v.10.4.1).
(TIF)

**S5 Fig. RT-PCR confirmation that *s*Gff resides in the spermathecae of mated, *s*Gff⁺ females.** Semi-quantitative RT-PCR analysis to confirm that live *s*Gff cells reside in the spermathecae of mated, *s*Gff⁺ females but not their *s*Gff⁻ counterparts. *Gff β-tubulin* was used as a loading control. Each of the three *s*Gff⁺ *and s*Gff⁻ samples was derived from spermathecae extracted and pooled from three individual flies.
(TIF)

**S6 Fig. Relative *Gmmcpt1* transcript abundance in spermathecae of 10 day old virgin and mated *Gmm* females.** Mating significantly increases *Gmmcpt1* expression (an average of 7.5x) in the spermatheca of *Gmm* females. Each dot on the graph represents one biological replicate (each replicate including spermathecae from three individuals). Statistical significance was determined via student's t-test (GraphPad Prism v.10.4.1).
(TIF)

**S7 Fig. Efficiency of RNAi mediated knockdown of *cpt1* expression in spermathecae and testes of mated female and male *Gmm*.** Relative *Gmmcpt1* transcript abundance in dsCPT1 (treatment) and dsGFP (control) treated *Gmm* females (10 day old mated) and males (seven days old). RNAi reduced *cpt1* expression in spermathecae and testes by an average of 66% and 56%, respectively. Each dot on the graph represents one biological replicate (each including spermathecae and testes from three mated, dsRNA treated flies). Statistical significance was determined via student's t-test (GraphPad Prism v.10.4.1).
(TIF)

**S8 Fig. Treatment of mated *Gff* females with etomoxir, an irreversible inhibitor of CPT1, inhibits the motility of stored spermatozoa.** (A) Motility, as a measure of flagellar beat frequency in hertz (Hz), of spermatozoa housed in the spermathecae of mated 10 day old treatment (etomoxir) and control (PBS) *Gff* females. Points on the graph represent one biological replicate (one replicate equals the average beat frequency of two sperm tails from one spermatheca). Bars represent median values, and statistical significance was determined via Student's t-test (GraphPad Prism v.10.4.1). (B) Etomoxir LD$_{50}$ dose response curve. The effective dose at which 50% of treated flies perished was 90 µM of etomoxir. As such 50 µM doses were used to quantify the impact of etomoxir mediated CPT1 inhibition on the motility of stored tsetse sperm.
(TIF)

**S9 Fig. *Wigglesworthia* and *Sodalis* density in mated dsCPT1(treatment) compared to dsGFP (control) treated Gmm at the end of their 4ᵗʰ GC.** RT-qPCR was used to determine that RNAi-mediated knockdown of *Gmmcpt1* did not

detrimentally impact obligate *Wigglesworthia*. While this treatment did significantly decrease *Sodalis* density, this bacterium, unlike *Wigglesworthia*, has no documented impact on tsetse fecundity.
(TIF)

**S10 Fig. Acylcarnitines are necessary for spermatozoa to remain viable following copulation, as evidenced by the fact that the spermathecae of mated *Gmm* female flies (which do not harbor *Spiroplasma*) produce more acylcarnitines than do those of their virgin counterparts.** Additionally, experimental inhibition of the carnitine shuttle via RNAi mediated knockdown of *carnitine O-palmitoyltransferase-1* (dsCPT1), and thus acylcarnitine production, reduces the motility of stored sperm, which are eventually resorbed thus resulting in reproductive sterility. SP, spermatheca; FB, fat body; HL, hemolymph; GC, gonotrophic cycle.
(TIF)

**S11 Fig. Example gel indicating the *s*Gff infection status of all *Gff* flies used in this study.** Genomic DNA extracted from legs (lab reared flies) or the terminal end of reproductive organ-containing abdominal tissue (field captured flies) of adult female and male *Gff* was subjected to PCR analysis using primers that specifically amplify *Gff β-tubulin* and *s*Gff *16S rRNA*. One microliter of PCR product from all samples was loaded onto the gel.
(TIF)

## Acknowledgments

We are grateful to Sidiya Mbodj (Department of Epidemiology of Microbial Diseases, Yale School of Public Health) for rearing the *Gmm* used in this study. We would like to thank all involved personnel at the Joint IAEA-Insect Pest Control Laboratory in Seibersdorf, Austria, for rearing *Gff* and providing pupae for this work. We thank Dr. Paul Mireji (Biotechnology Research Institute-Kenya Agricultural and Livestock Research Organization) for assistance with the collection of all wild tsetse flies used in this study.

## Author contributions

**Conceptualization:** Brian L Weiss, Geoffrey M. Attardo, Francesca Scolari, Daniel J. Bruzesse, Serap Aksoy.

**Data curation:** Brian L Weiss, Geoffrey M. Attardo, Robert T. Koch.

**Formal analysis:** Brian L Weiss, Fabian Gstöttenmayer, Geoffrey M. Attardo.

**Funding acquisition:** Brian L Weiss, Francesca Scolari, Serap Aksoy.

**Investigation:** Brian L Weiss, Fabian Gstöttenmayer, Erick Awuoche, Gretchen M. Smallenberger, Geoffrey M. Attardo, Francesca Scolari, Robert T. Koch, Richard Echodu, Robert Opiro.

**Methodology:** Brian L Weiss, Fabian Gstöttenmayer, Geoffrey M. Attardo, Francesca Scolari.

**Project administration:** Brian L Weiss, Francesca Scolari.

**Resources:** Brian L Weiss, Fabian Gstöttenmayer, Francesca Scolari, Richard Echodu, Robert Opiro, Anna Malacrida, Adly MM Abd-Alla, Serap Aksoy.

**Software:** Geoffrey M. Attardo.

**Supervision:** Brian L Weiss.

**Validation:** Brian L Weiss.

**Visualization:** Brian L Weiss, Fabian Gstöttenmayer, Geoffrey M. Attardo.

**Writing – original draft:** Brian L Weiss, Francesca Scolari, Adly MM Abd-Alla, Serap Aksoy.

**Writing – review & editing:** Brian L Weiss.

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
