## [Decision Letter · Decision Letter 0]

12 Nov 2025

PGENETICS-D-25-01020

Endosymbiont hijacking of acylcarnitines regulates insect vector fecundity by suppressing the viability of stored sperm

PLOS Genetics

Dear Dr. Weiss,

Thank you for submitting your manuscript to PLOS Genetics. After careful consideration, we feel that it has merit but does not fully meet PLOS Genetics's publication criteria as it currently stands. Therefore, we invite you to submit a revised version of the manuscript that addresses the points raised during the review process.

Please submit your revised manuscript within by Dec 12 2025 11:59PM. If you will need more time than this to complete your revisions, please reply to this message or contact the journal office at plosgenetics@plos.org. Please include the following items when submitting your revised manuscript:

We look forward to receiving your revised manuscript.

Kind regards,

Kevin J Vogel, Ph.D.

Guest Editor

PLOS Genetics

Kelly Dyer

Section Editor

PLOS Genetics

Aimée Dudley

Editor-in-Chief

PLOS Genetics

Anne Goriely

Editor-in-Chief

PLOS Genetics

**Additional Editor Comments:**

Three reviewers have provided feedback on your manuscript "Endosymbiont hijacking of acylcarnitines regulates insect vector fecundity by suppressing the viability of stored sperm." All three are in agreement that the study is well performed, described clearly, and is a significant contribution to our understanding of host-symbiont interactions. They each provide comments to clarify and improve the manuscript that should be addressed in your resubmisision.

**Journal Requirements:**

At this stage, the following Authors/Authors require contributions: Brian L Weiss. Please ensure that the full contributions of each author are acknowledged in the "Add/Edit/Remove Authors" section of our submission form.

The list of CRediT author contributions may be found here: https://journals.plos.org/plosgenetics/s/authorship#loc-author-contributions

- © on page: 30

- ® on page: 34

- TM on page: 34.

Potential Copyright Issues:

i) Figure 5. Please confirm whether you drew the images / clip-art within the figure panels by hand. If you did not draw the images, please provide (a) a link to the source of the images or icons and their license / terms of use; or (b) written permission from the copyright holder to publish the images or icons under our CC BY 4.0 license. Alternatively, you may replace the images with open source alternatives. See these open source resources you may use to replace images / clip-art:

**Reviewers' comments:**

Reviewer's Responses to Questions

**Comments to the Authors:**

Reviewer #1: The bases of most bacterial symbioses in insects are poorly understood mechanistically. This is a beautiful study exploring a possible cause of competition between spiroplasmas and the tsetse fly. The methods are clearly described and robust. The inclusion of field collected tsetse’s is a nice addition, even though they cannot be age controlled. The strength of the effects are strong and believable – sperm motility, increase in acylcarnitines despite overall reduction in lipids, reduction in spiroplasmas following RNAi of acylcarnitine synthesis gene, and patterns of gene expression that correlate with spiroplasma infection, mating, and beat frequency. The assembly of experimental approaches gives weight to the authorr's arguments. The discussion is thoughtful and incorporates relevant data across systems to provide support for speculative intepretation.

This paper was a joy to read!

Minor

Line 67 awkward. Reword.

Line 92 ‘they’ refers to trypanosomes. Confusing. Reword.

Line 175. Remind reader why lipidomics rather than a metabolomics approach

Reviewer #2: Review uploaded as an attachment.

Reviewer #3: The manuscript titled “Endosymbiont hijacking of acylcarnitines regulates insect vector fecundity by suppressing the viability of stored sperm” describes a solid study investigating how the parasitic bacteria Spiroplasmsa glossinidia (sGff) impacts the viability of tsetse fly sperm stored in the female spermathecae (sperm storing organ). The authors show that sGff impacts motility in sperm stored in the female spermathecae, but not the male testes and use RNAi and gene expression analysis to provide mechanistic evidence that the impact on spermathecae stored sperm is mediated by fly-produced acylcarnitines. Overall, this is a clean study that presents an advancement in our understanding how Spiroplasma impacts tsetse fly fecundity. I have a few minor suggestions and comments that should be considered to improve the manuscript.

Suggestions and comments

Remove speculative statements beginning with “likely,” such as “likely other tsetse species as well” (Line 153), from the Results section.

Lines 186–188: The statement “spermatozoa stored in the spermathecae of sGff+ females exhibit a motility defect” should be revised. Because sperm in male testes are inherently less active, and sperm are more active in spermathecae regardless of female infection status, it is not possible to distinguish a motility defect from an activation defect. The phrasing should remain agnostic to mechanism.

Lines 188–190: The claim that “mated Gff females that housed the bacterium presented with a relatively increased abundance of acylcarnitines in their FB” requires a reference (either to a publication or figure). The same applies to Lines 243–244: “We observed that the FB of mated sGff+ females produces more acylcarnitines than does that of their sGff– counterparts.”

What is the baseline lipidome of unmated females with and without Spiroplasma? If infection already alters lipid profiles before mating, this could affect interpretation of how sGff influences lipids in mated females.

Where multiple t-tests were performed (e.g., Line 225), please clarify whether a multiple-testing correction was applied.

The acronyms “sGff” and “Gff” are very similar and sometimes confusing. Consider using clearer or more distinct identifiers, as the similarity occasionally obscures meaning.

Line 245: The statement “This finding suggests that both hemolymph-borne and sGff localized within fly tissues may be metabolizing acylcarnitines” is not directly supported by the data. If retained, the reasoning should be explicitly stated; otherwise, it should be removed.

Several parenthetical asides are too informal for a manuscript, such as Line 266: “[amongst other things, see Materials and Methods…].” These should be rephrased or omitted.

The manuscript alternates between metabolite measurements in hemolymph and gene expression perturbations in spermathecae. Although this design is reasonable, the authors should acknowledge that these are distinct tissues and that conclusions about their interaction are inferred. Because Gff can inhabit the gut, this could also influence hemolymph lipid changes. Likewise, etomoxir treatment affects multiple tissues, so a more holistic discussion of its systemic effects—both within and outside the spermathecae—would strengthen the interpretation.

The abbreviation “GC” should be reintroduced when it reappears in the Results, as it was easy to forget by that point.

The reduced Sodalis loads observed in dsCPT1 flies merit more discussion. Although the authors state that Sodalis “has no known function related to tsetse reproduction,” absence of evidence is not evidence of absence. This finding should be interpreted rather than dismissed.

It would be helpful to mention earlier that sGff cannot synthesize lipids de novo, as this is an important aspect of its metabolism.

An additional consideration is that sGff-infected females might need to mate more frequently to maintain fecundity, potentially providing a transmission advantage to sGff. This possibility is worth noting.

Lines 526–527: The statement “we observed glandular cells associated with the outer perimeter of tsetse’s spermathecal reservoir” should clarify whether this is a novel observation or one previously reported.

Finally, please remove asterisks from lipidome panels where compounds only trend toward a difference but are not statistically significant. Because asterisks conventionally denote significance, an alternate marking would be more appropriate.

**Have all data underlying the figures and results presented in the manuscript been provided?**

Reviewer #1: Yes

Reviewer #2: Yes

Reviewer #3: Yes

PLOS authors have the option to publish the peer review history of their article (what does this mean? ). If published, this will include your full peer review and any attached files.

**Do you want your identity to be public for this peer review?** For information about this choice, including consent withdrawal, please see our Privacy Policy .

Reviewer #1: No

Reviewer #2: No

Reviewer #3: No

**Figure resubmission:**

**Reproducibility:**



---

## [Editor Report · Decision Letter 1]

1 Dec 2025

Dear Dr Weiss,

We are pleased to inform you that your manuscript entitled "Endosymbiont hijacking of acylcarnitines regulates insect vector fecundity by suppressing the viability of stored sperm" has been editorially accepted for publication in PLOS Genetics. Congratulations!

Yours sincerely,

Kevin J Vogel, Ph.D.

Guest Editor

PLOS Genetics

Kelly Dyer

Section Editor

PLOS Genetics

Aimée Dudley

Editor-in-Chief

PLOS Genetics

Anne Goriely

Editor-in-Chief

PLOS Genetics

BlueSky: @plos.bsky.social

Comments from the reviewers (if applicable):

**Data Deposition**

http://datadryad.org/submit?journalID=pgenetics&manu=PGENETICS-D-25-01020R1

**Press Queries**

---

## [Editor Report · Acceptance letter]

PGENETICS-D-25-01020R1

Endosymbiont hijacking of acylcarnitines regulates insect vector fecundity by suppressing the viability of stored sperm

Dear Dr Weiss,

We are pleased to inform you that your manuscript entitled "Endosymbiont hijacking of acylcarnitines regulates insect vector fecundity by suppressing the viability of stored sperm" has been formally accepted for publication in PLOS Genetics! Your manuscript is now with our production department and you will be notified of the publication date in due course.

With kind regards,

Anita Estes

PLOS Genetics

On behalf of:
